# 9-fold symmetry is not essential for centriole elongation and formation of new centriole-like structures

**Pallavi Panda** [1], **Mark S. Ladinsky** [1] **& David M. Glover** [1] ✉

As daughter centrioles assemble during G2, they recruit conserved Ana3/RTTN followed by its partner Rcd4/PPP1R35. Together, this contributes to the subsequent recruitment of Ana1/CEP295, required for the centriole's conversion to a centrosome. Here, we show that Rcd4/PPP1R35 is also required to maintain 9-fold centriole symmetry in the *Drosophila* male germline; its absence causes microtubule triplets to disperse into a reduced number of doublet or singlet microtubules. *rcd4*-null mutant spermatocytes display skinny centrioles that elongate normally and localize centriolar components correctly. Mutant spermatocytes also have centrioles of normal girth that splay at their proximal ends when induced to elongate by Ana1 overexpression. Skinny and splayed spermatid centrioles can still recruit a proximal centriole-like (PCL) structure marking a capability to initiate features of centriole duplication in developing sperm. Thus, stable 9-fold symmetry of microtubule triplets is not essential for centriole growth, correct longitudinal association of centriole components, and aspects of centriole duplication.

Centrioles are the core components of centrosomes and have a double life as the basal bodies of cilia and flagella[1]. Their duplication is initiated by the phosphorylation of Ana2 (*Drosophila*)/ STIL (human) which enables the assembly of Sas6 into a cartwheel structure in G1/S phase of the cell cycle[2–6]. The subsequent recruitment of Sas4/CPAP promotes polymerization of centriolar microtubules[7–10]. The Ana1/Cep295 protein is recruited to the daughter centriole as cells pass through mitosis enabling the daughter to recruit Asterless (Asl)/Cep152 and so be competent to recruit pericentriolar material and become a centrosome[11–13]. We have previously shown that both Cep135 and the Ana3:Rcd4 complex are required for the recruitment of Ana1[13,14]. Our previous study described an *rcd4[1]* deletion, which leaves an intact start codon in-frame with the conserved C-terminal 68 amino acids that have residual function, whereas the entire open reading frame is deleted in the *rcd4[2]* null allele[14]. The somatic cells of both of these *rcd4*-mutant flies have fewer centrioles, aberrant mitoses, and a reduced number of basal bodies in the femoral chordotonal sensory organs leading to loss of coordination of adult flies[14]. Moreover, embryos derived from *rcd4*-mutant mothers were unable to develop because of mitotic defects resulting from the loss of centrioles. However, when

the function of *rcd4* was restored in the nervous system to rescue coordination, this enabled both *rcd4[1]* and *rcd4[2]* mutant males to mate and generate progeny indicating that Rcd4 was not absolutely essential for male germline function. We described how primary spermatocytes of *rcd4[1]* males contained the wild-type complement of two pairs of centrioles whereas *rcd4[2]* mutant spermatocytes exhibited around 29% centriole loss[14].

In this work, we have sought to examine the organization of the residual centrioles in *rcd4[2]* mutant males. We report a male germline-specific role for Rcd4 in maintaining the stability and cohesion of microtubule triplets. Importantly, we show that centriole ninefold microtubule symmetry is not essential for centriole elongation, correct distribution of the centriole's components along its length, and in initiating aspects of centriole duplication.

## Results

### *rcd4*-Skinny centrioles lack 9-fold microtubule triplet symmetry

We first confirmed the number of centrioles in spermatocyte cysts of both *rcd4*-mutant males. Immunostaining revealed that 16-cell cysts of *rcd4[1]* primary spermatocytes have 64 centrioles (32 pairs), with Asl

[1]Division of Biology and Biological Engineering, California Institute of Technology, Pasadena, CA 91125, USA. ✉e-mail: dmglover@caltech.edu

along their lengths and Drosophila Pericentrin-like protein (D-Plp) at their proximal region, that appear indistinguishable from wild-type spermatocyte cysts (Fig. 1a, b and Supplementary Fig. 1). However, we observed that in addition to the 40.7% of centriole loss observed in *rcd4²* spermatocytes (N = 137), 33.2% of the remaining centrioles were paired and had seemingly normal appearance, and 13.8% were seemingly normal unpaired single centrioles (Fig. 1a-c). In addition, we also observed "skinny" centrioles that were similar in length to ones of normal appearance with Asl along their lengths and D-Plp at their presumptive proximal ends. The skinny centrioles were paired with a centriole of apparently normal morphology in 6.4% of cells or were unpaired in 5.9% of cells. (Fig. 1b, c).

Optical sectioning revealed parallel lines of Asl staining representing the two opposing sides of the Asl cylinder in wild-type centrioles. In contrast, the skinny centrioles had a single line of Asl staining suggesting the absence of part of the centriole wall extending along the entire centriole's length (Fig. 1b). To determine how much of the centriole remained, we carried out electron tomography on preparations of *rcd4²* mutant testes. This revealed the skinny centrioles to comprise two or three closely associated microtubules that could either lie in proximity to centrioles of wild-type appearance (paired skinny centriole) or be free within the cytoplasm (unpaired skinny centriole) (Fig. 1d). The pairing of a skinny centriole orthogonal to one of wild-type appearance suggests that the former is not derived by breakdown of the latter but that it arose independently in the current or previous cell generation. Centriole breakdown resulting from mutation in *rcd4* appears to result in extreme destablization of the lateral interactions between triplet microtubules that normally result in the characteristic ninefold symmetrical arrangement of microtubules.

### rcd4-Skinny centrioles undergo elongation

The presence of *rcd4²* spermatocytes lacking centrioles, having just one centriole pair, or having single centrioles suggests that in addition to the structural defects underlying the formation of skinny centrioles, there must also be direct or indirect consequences for the centriole duplication cycle in the four spermatogonial divisions that generate the cyst of 16 primary spermatocytes (Fig. 2a). Once the 16-cell cyst has been generated, the spermatocytes undertake a prolonged G2 phase of around 72 h during which there is extensive cell growth accompanied by elongation of the centrioles. The mature primary spermatocytes then undertake two rounds of meiotic divisions to generate cysts of 64 spermatids each having a single centriole in the wildtype (Fig. 2a).

Each wild-type spermatogonium enters mitosis with two pairs of centrioles and contributes two mature centrioles to each one of its daughter cells. Procentrioles will develop orthogonally to these mature centrioles in the G1 phase of their next cycle. Examples of wild-type spermatogonia in G1 are shown in Fig. 2b. Within each cell can be seen two rings of Asl staining corresponding to the walls of the mature centrioles segregated in the previous mitosis. In contrast, spermatogonia at a corresponding stage from *rcd4²* mutant testes displayed a similar arrangement of centrioles as we had seen in spermatocytes: some cells had no centrioles and others had centriole fragments that correspond to the skinny spermatocyte centrioles. The skinny spermatogonial centrioles could exist either singly, as pairs, or paired with centriole of seemingly normal appearance (see the enlarged image of centrioles from a *rcd4²* spermatagonium in Fig. 2c). An equivalent set of structures were observed in early primary spermatocyte cysts although these were shorter than those seen towards the end of the extended G2 phase (Fig. 2b, c). Thus, the structural consequences of the absence of Rcd4 protein are manifested in the spermatogonial divisions as centriole loss and generation of skinny centrioles giving rise to spermatocytes having normal and skinny centrioles. Together with centrioles of normal appearance, skinny centrioles are able to

undergo elongation in the spermatocyte's extended G2 phase and through the meiotic divisions before segregating into spermatids (Fig. 2c). To assess whether both normal and skinny centrioles could undergo elongation at comparable rates in G2 spermatocytes, we captured images of *rcd4²* paired skinny centrioles in various stages of their elongation and compared the length of these two centriole types. Immunofluorescence images of such centrioles stained to reveal both Asl and D-Plp suggested that skinny centrioles did indeed elongate in concert with centrioles of normal appearance (Fig. 2d). Thus, the rate and extent of elongation of skinny centrioles is equivalent to centrioles of normal appearance.

### Centriole components distribute correctly along rcd4-skinny centrioles

We have previously shown that the complex formed between Rcd4 and Ana3 is required in somatic cells to efficiently recruit the Ana1 protein to daughter centrioles enabling them to recruit Asl and components of the pericentriolar material, in progression through mitosis[14]. This led us to ask whether specific centriole components might be absent from centrioles in *rcd4²* mutant primary spermatocytes. We have previously defined a core region (zone I) of the *Drosophila* somatic centriole occupied by the cartwheel proteins that is surrounded by annuli of increasing radii that we termed zones II and III[15]. The male germline centrioles have analogous regions (see schematic in Fig. 3a). Here, we found that the C-terminal part of Ana1 and the N-terminal part of Asl were present in zones II and III respectively and extended along the lengths of *rcd4²* spermatocyte centrioles of normal and paired skinny appearance in agreement with previous reports of wild-type centriole organization[14–16] (Fig. 3b). Moreover, Cep135 was also present in these structures in *rcd4²* spermatocytes and was enriched at the proximal end (Fig. 3b). Thus, it seems likely that the previously described Cep135, Ana1, Asl network that extends from zone I to zone III remains intact in the absence of Rcd4. Similarly, we found that the cartwheel components Sas6 and Gorab were within zone I, Sas4 was in Zone II, and D-Plp was in zone III, at the proximal ends of the *rcd4²* spermatocyte centrioles of normal and paired skinny appearance (Fig. 3b). Strikingly, each of these proteins was also associated with unpaired skinny centrioles at an appropriate position along the centriole's length (Fig. 3b). As the ninefold symmetry of microtubule blades is lost in skinny centrioles, this indicates that such symmetry is not necessary to maintain the interactions of proximally located cartwheel components or of proteins associated with the centriole walls and lumen along the centriole's length.

The only centriole component that we were unable to detect on either the normal or skinny centrioles in the *rcd4²* spermatocytes was Rcd4's partner protein, Ana3 (Fig. 3b). This was a surprising finding since our study of somatic centrioles had previously indicated that Rcd4 was loaded onto centrioles after Ana3 and that Rcd4 recruitment was in fact dependent on Ana3, and not vice versa. This led us to examine the association of Ana3 in *rcd4²* spermatogonial and spermatocyte centrioles at different stages of their growth in G2 (Fig. 3c). This revealed that Ana3 was associated with centrioles in *rcd4²* spermatogonia and in newly formed primary spermatocytes, albeit asymmetrically between the duplicated centriole pair, but Ana3 was not present in primary spermatocytes that had commenced centriole elongation (Fig. 3c). Examination of spermatogonia and spermatocytes in *rcd4¹* hypomorphic males also revealed loss of Ana3 in spermatocytes undergoing centriole elongation (Supplementary Fig. 2). We therefore conclude that Rcd4 is not required for Ana3 recruitment but for its maintenance during centriole elongation.

### rcd4 centrioles of normal girth splay upon elongation

Although *rcd4²* spermatocyte cysts contained a proportion of centrioles of apparently normal appearance, we wondered whether

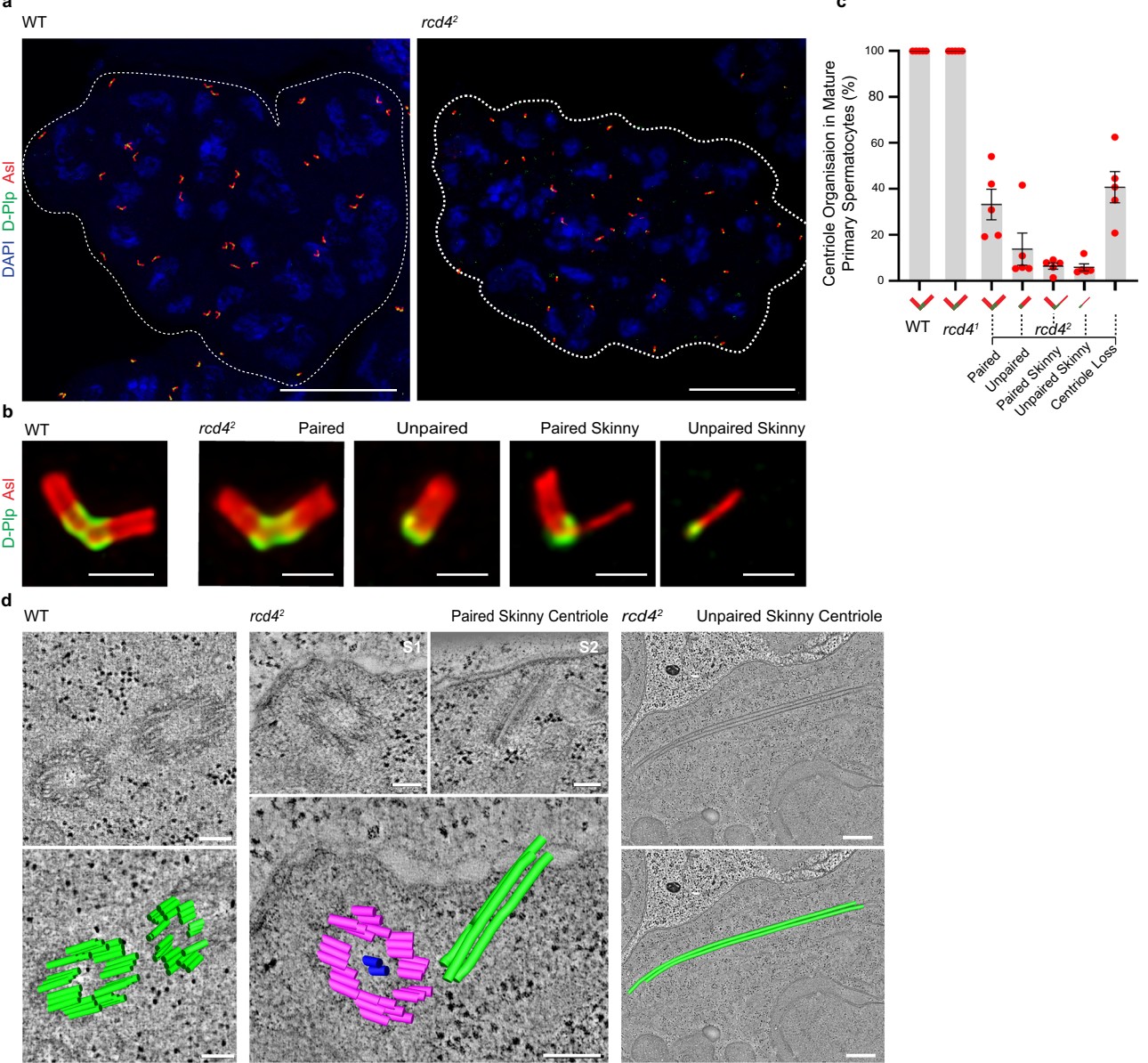

**Fig. 1 | *rcd4*-Null mutant spermatocytes have structurally abnormal skinny centrioles comprising two or three microtubules. a** Mature primary spermatocyte cysts of wildtype (WT) and *rcd4²* males immuno-stained to reveal elongated centrioles; Asl (red), D-Plp (green), and DNA (blue). The boundary of the 16-cell cyst is outlined with dotted lines. Scale bar, 30 μm. **b** Magnified examples of elongated centrioles from mature primary spermatocytes of WT and *rcd4²* males immuno-stained to reveal Asl (red) and D-Plp (green). Scale bar, 1 μm. **c** Percentage of centrioles in mature primary spermatocytes of WT, *rcd4¹* and *rcd4²* males that fall into the following categories: paired, unpaired, paired skinny, unpaired skinny centrioles and centriole loss. Note that centrioles in WT and *rcd4¹* primary spermatocytes are all paired in the typical V-shape configuration (mean = 100%) and hence other categories are not represented graphically. Mean percentages of *rcd4²* primary spermatocyte centrioles in each category: paired (33.2%), unpaired (13.8%),

paired skinny (6.4%), unpaired skinny (5.9%) and overall centriole loss (40.70%). Mean values and Standard Deviations (SDs) are shown for five independent testes analyzed per genotype. $N = 140$ mature primary spermatocytes scored for each genotype with $n = 20{-}37$ spermatocytes scored per testis. Source data are provided as a Source Data file. **d** Electron tomography of WT and *rcd4²* primary spermatocytes revealing the ultra-structure of paired skinny and unpaired skinny centrioles. A selected single section and a two-dimensional model overlay are illustrated. *rcd4²* paired skinny centriole: S1 and S2 are individual sections representing the two centrioles in the pair. Refer to Supplementary Movies 1 and 2 for entire z-stack volume. Scale bar, 100 nm (WT and *rcd4²* paired skinny centriole) and 500 nm (unpaired skinny centriole). The experiment was repeated three times with similar results.

---

increasing the stress on these structures might reveal structural defects. To address this, we chose to overexpress Ana1, which has previously been shown to lead to centriole elongation[16]. We confirmed that expression of GFP-tagged Ana1 (GFP-Ana1) from a poly-Ubiquitin promoter (*Ana1^OE*) led to an increase in the length of otherwise wild-type centrioles from 1.416 (+/− 0.0541) μm to 2.20 (+/− 0.773) μm (Supplementary Fig. 3). When Ana1 was overexpressed from the same promoter in an *rcd4¹* or *rcd4²* mutant background, we found that

approximately 19.1% and 62.2% of centrioles of normal girth, respectively, became dramatically splayed at one end (Fig. 4a-c).

To examine the ultrastructural arrangement of microtubules in the splayed centrioles, we carried out electron tomography. This revealed a range of structures displaying restricted to extensive degrees of splaying. The *rcd4²;Ana1^OE* centriole exhibiting restricted splaying in Fig. 4d has nine sets of microtubule triplets extending along the entire length captured in this physical section. However,

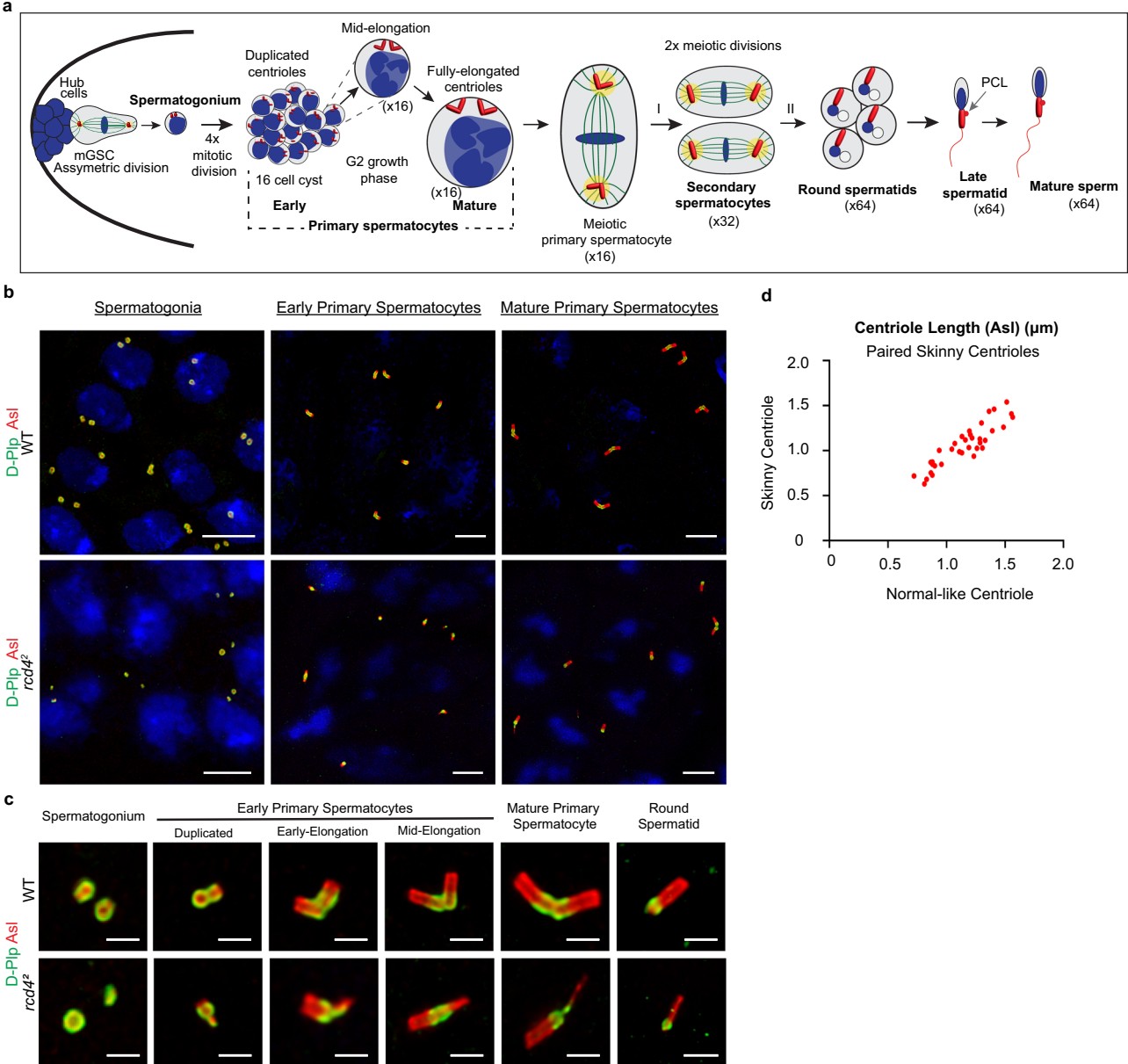

**Fig. 2 | Skinny centrioles in *rcd4²* primary spermatocytes elongate comparably to structurally normal centrioles during the extended G2 growth phase.**
**a** Schematic of centriole development in *Drosophila* spermatogenesis. The male germline stem cell (mGSC) undergoes asymmetric cell division giving rise to a renewed stem cell and a differentiating spermatogonium containing two centrioles. A spermatogonium undergoes four rounds of mitotic divisions resulting in a 16-cell cyst of early primary spermatocytes. The early primary spermatocytes pass through an extended G2 growth phase during which the two inherited centrioles in each cell duplicate and significantly grow in length alongside overall cell growth. The mature primary spermatocyte undertakes two meiotic divisions segregating the four elongated centrioles individually into round spermatids. The round spermatids undergo spermiogenesis developing into late spermatids, where the centriole/basal body recruits a proximal centriole-like (PCL) structure and templates the formation of the flagella/sperm tail, before differentiating into mature sperms. A single cell of each type has been represented for clarity. Schematic not drawn to scale. **b** WT and *rcd4²* spermatogonia, early and mature primary spermatocytes immuno-sained to reveal Asl (red), D-Plp (green) and DNA (blue). Scale bar, 5 μm. The experiment was repeated more than three times with similar results. **c** Representative WT and *rcd4²* centrioles in spermatogonia, early primary spermatocytes (centriole duplication, early elongation and mid-elongation stages), mature primary spermatocytes, and in round spermatids. Centrioles were stained to reveal Asl (red) and D-plp (green). Scale bar, 1 μm. **d** *rcd4²* paired skinny centrioles were analyzed to compare centriole elongation in skinny versus normal centrioles in the pair. Centriole length measurements were made using Asl as a centriole marker at various elongation stages in *rcd4²* mutant primary spermatocytes. Centriole lengths (in μm) of the skinny versus normal centrioles in paired skinny centrioles were plotted against each other. Each data point represents the respective skinny and normal centriole lengths. *N* = 35 such centriole pairs were analyzed. Source data are provided as a Source Data file.

while the triplets are in close proximity and physically connected to each other in the deep part of the tomogram, they gradually become separated and distanced from one another along the length of the centriole (Fig. 4d and Supplementary Movie 4,5). In the second example of an *rcd4²;Ana1^OE^* centriole exhibiting extensive splaying, the microtubule triplets are reduced to doublet or singlet microtubules (Fig. 4d). The centriole comprises 7 sets of doublet microtubules and two singlets. The singlet microtubules are splayed away from the main structure and are dramatically kinked (Fig. 4d and Supplementary Movie 6).

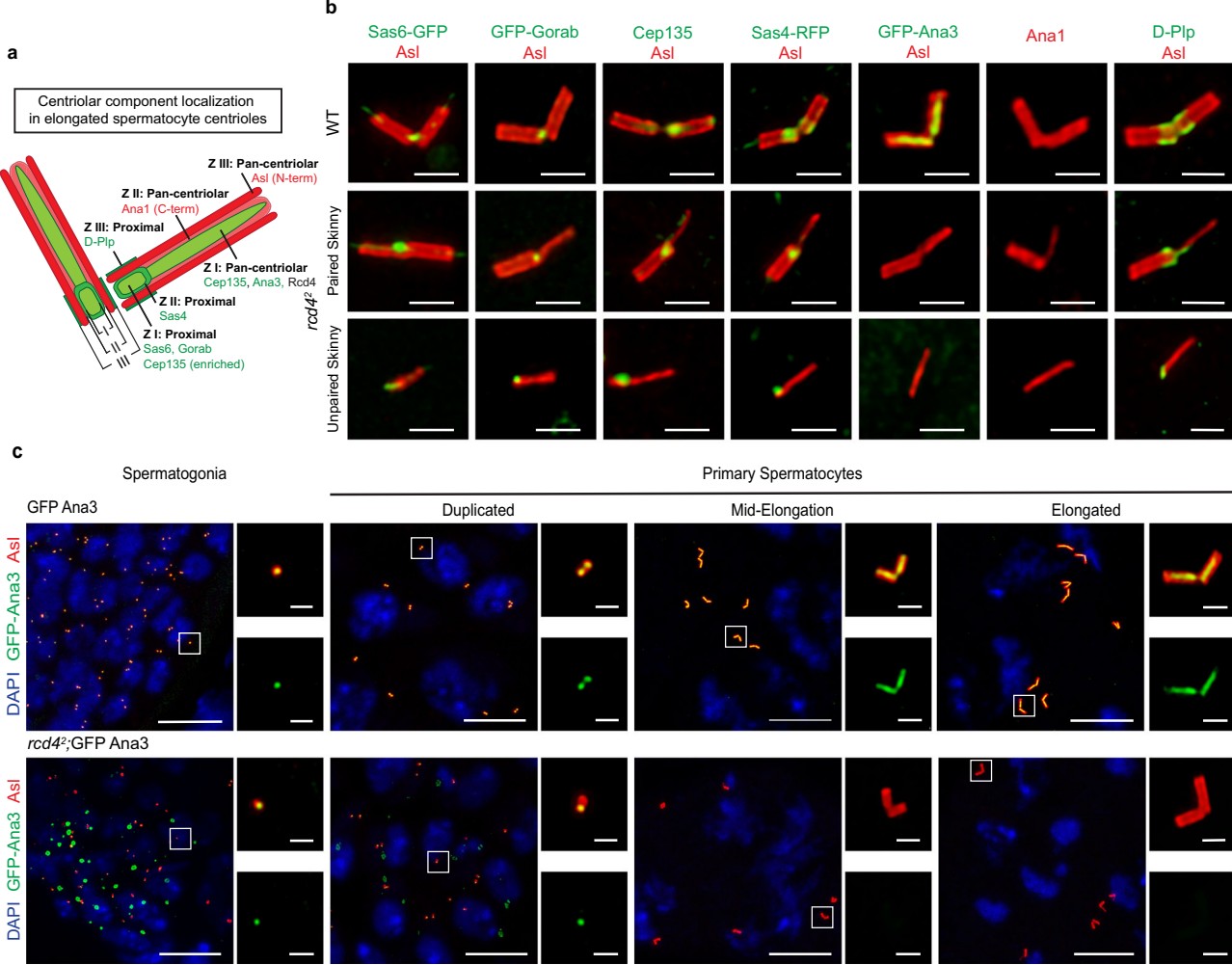

**Fig. 3 | *rcd4²* skinny centrioles appropriately localize key centriolar components along their lengths, except for Ana3, Rcd4's binding partner. a** Schematic of centriolar component localization in elongated spermatocyte centrioles. Centriolar zones I, II and III with respective components localized at the centriole proximal region or pan-centriolar, as illustrated in the schematic. Localizations depicted specifically for Ana1 (C-terminal) · zone II and Asl (N-terminal) – zone III. **b** Elongated WT and skinny *rcd4²* centrioles (paired and unpaired) were immunostained to reveal Asl (red), Cep135 (green), Ana1 (red) and D-Plp (green). Transgenic flies expressing poly-Ubiquitin driven Sas6-GFP, GFP-Gorab, Sas4-RFP and

GFP-Ana3 (all represented in green) were also used. Scale bar, 1 μm. The experiment was repeated three times with similar results. **c** GFP-Ana3 (control) and *rcd4²*;GFP-Ana3 male germline cells in spermatogonia and primary spermatocytes at centriole duplication, mid-elongation and elongated stages stained to reveal GFP-Ana3 (green), Asl (red) and DNA (blue). Note GFP-Ana3 aggregates in *rcd4²*; GFP-Ana3 spermatogonial and early primary spermatocyte cells. Scale bars, 10 μm (overview image) and 1 μm (insets). The experiment was repeated more than three times with similar results.

## Centriole splaying initiates at the proximal end of *rcd4* centrioles

We then examined how the splaying developed as centrioles elongated in primary spermatocytes and whether the splayed centrioles remained stable in haploid spermatids. We found that splaying did not become evident until the late stages of centriole elongation when the centrioles were nearly fully elongated (Fig. 5a). When we further examined the individual centrioles segregated into spermatids following the two meiotic divisions, we found splaying to be even more prominent, often spreading throughout the length of the centriole (Fig. 5a). The extent of splaying, measured as the angle between the two furthermost ends of the splayed centriole, averaged at approximately 27.2° but also reached as much as 84.7° in some extreme cases (Fig. 5b). The snapshot examples in Fig. 5c reveal intermediate steps of *rcd4²* centrioles of normal grith loosing microtubules in the process splaying.

To determine which end of the centriole was exhibiting splaying we co-stained the structures to reveal the proximal end markers Sas6 and D-Plp. This revealed that centriole splaying was initiated from the proximal end of the centriole and that whereas D-Plp always associated

with all splayed ends, Sas6 could associate with a few or multiple ends (Fig. 5d).

## Skinny and splayed *rcd4* centrioles initiate centriole-like duplication

As the *Drosophila* spermatid begins to elongate, centriole duplication factors are recruited to a point close to the base of the main giant centriole to form an atypical centriole known as the proximal centriole-like (PCL) structure[17]. The PCL has features of daughter centrioles: it forms near the proximal part of the giant centriole; it contains centriolar proteins Sas-6, Ana1, and Bld10/Cep135; and its formation requires Plk4 and Sas-6[17]. It lacks Sas4 and centriolar microtubules but is enriched in Poc1 and otherwise resembles an intermediate structure of a nascent centriole, and together with the giant centriole, it is able to mediate centrosome formation for the zygotic spindle and then undertake subsequent rounds of centriole duplication in the embryo[18].

The presence of PCL bodies near the proximal ends of both splayed and skinny centrioles in *rcd4²;Ana1^{OE}* spermatids (Fig. 6a) led

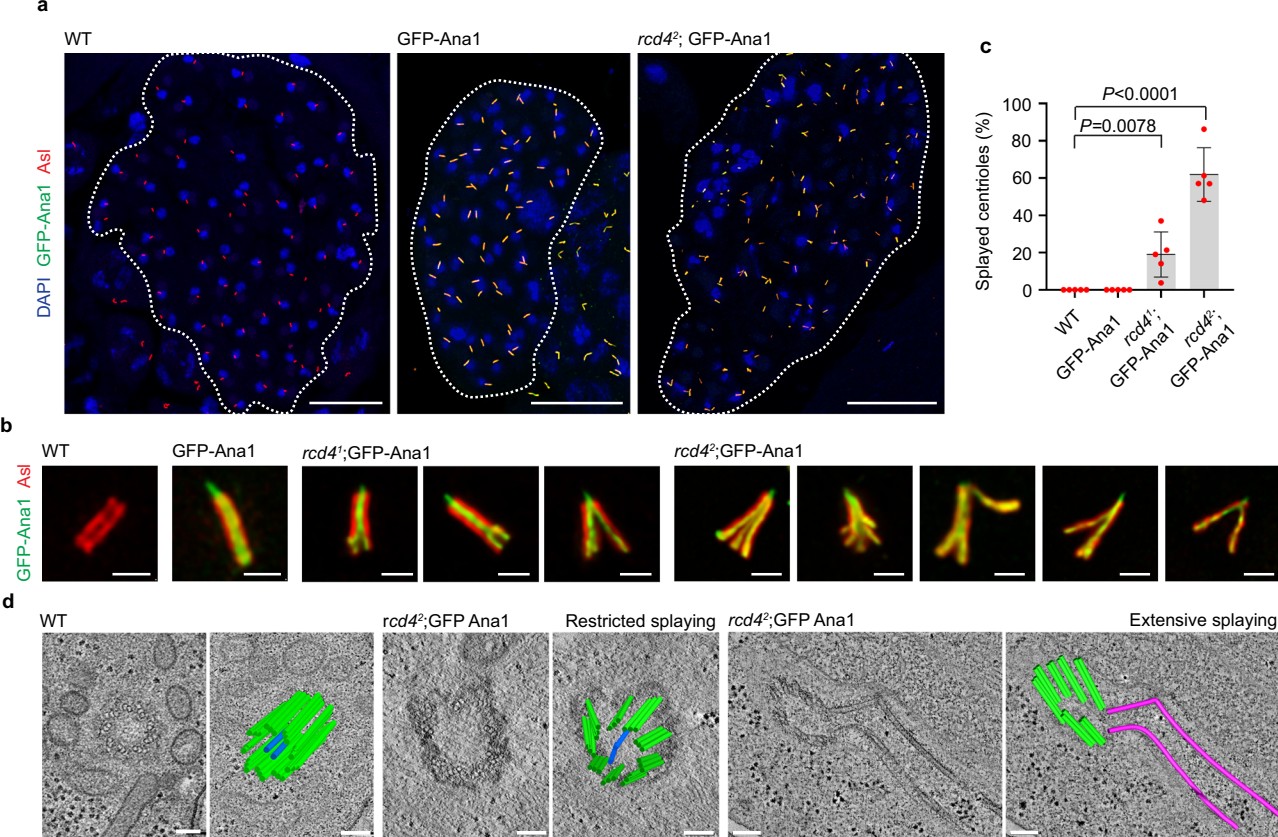

**Fig. 4 | Ana1 overexpression-driven centriole elongation results in the splaying of *rcd4*-mutant spermatid centrioles. a** Round spermatid cysts of WT, poly-Ubiquitin driven GFP-Ana1 overexpression (*Ana1^OE*) and *rcd4²*; GFP-Ana1 (*rcd4²*; *Ana1^OE*) males immuno-stained to reveal elongated centrioles with GFP-Ana1 (green), Asl (red), and DNA (blue). The boundary of the 64-cell cyst is outlined in dotted lines. Scale bar, 30 μm. **b** Magnified examples of round spermatid centrioles in WT and GFP-Ana1 overexpressing males, and splayed centrioles of *rcd4¹*;GFP-Ana1 and *rcd4²*;GFP-Ana1 males immuno-stained to reveal Asl (red) and GFP-Ana1 (green). Scale bar, 1 μm. **c** Quantification of the percentage of splayed centrioles observed in WT, GFP-Ana1, *rcd4¹*;GFP-Ana1 and *rcd4²*;GFP-Ana1 round spermatids. For *rcd4²*;GFP-Ana1, only non-skinny centrioles were taken into consideration for the quantifications. A two-tailed, unpaired *t* test was performed and

the *p*-values for *rcd4¹*;GFP-Ana1 and *rcd4²*;GFP-Ana1 are *P* = 0.0078 and *P* < 0.0001, respectively. Mean values for the percentage of splayed centrioles in *rcd4¹*;GFP-Ana1 and *rcd4²*;GFP-Ana1 are 19.1% and 62.2%, respectively. Mean values and SDs are shown for five independent testes per genotype. N = 250 spermatids analyzed for WT and GFP-Ana1; N = 289 spermatids analyzed for *rcd4¹*;GFP-Ana1 and N = 261 spermatids analyzed for *rcd4²*;GFP-Ana1. n = ~50 spermatids were scored per testis. Source data are provided as a Source Data file. **d** Electron tomography of WT and *rcd4²*;GFP-Ana1 spermatids revealing the ultra-structure of splayed centrioles. A single section and a two-dimensional model overlay are illustrated. Refer to Supplementary Movies 3, 4, 5 and 6 for the entire z-stack volume. Scale bar, 100 nm. The experiment was repeated more than three times with similar results.

us to relate PCL formation to nuclear morphology, which changes as spermatids elongate (Fig. 6b; Supplementary Fig. 4). In control *Ana1^OE* spermatids, none of the centrioles were splayed at the round spermatid stage and they all lacked PCLs. PCLs appeared synchronously within the spermatid cyst at the leaf nuclear stage as evident by the GFP-Ana1 positive bulge in close proximity to one end of all centrioles. Distinct individualized PCLs were then seen within a cyst of spermatids at the needle-like stage (Fig. 6b; Supplementary Fig. 4). The proportion of splayed centrioles in *rcd4²*;*Ana1^OE* spermatids showing extensive splaying at the round nuclear stage was 59.4% (*n* = 41/69) whereas splaying was restricted in 40.6% (*n* = 28/69) of the remaining splayed centrioles (Fig. 6c). None of these had an associated PCL indicating that the greater proportion of extensive centriole splaying takes place before PCL formation (Supplementary Fig. 4). The proportion of extensively splayed centrioles further increased to 68.9% (*n* = 42/61) at the leaf-shaped nuclear stage when single PCLs appeared simultaneously on all centrioles. We also observed the formation of the PCL on skinny centrioles at these stages. Thus, PCLs can assemble on both skinny and splayed centrioles that are losing the connections between the ninefold symmetrical microtubules.

This led us to ask the extent to which, when present, ninefold symmetrical units at the splayed proximal end of the centriole participated in the initiation of this form of centriole-like duplication to generate the PCL. At this stage of development, where we are currently not able to visualize the recruitment of Plk4 or Ana2, the molecules that initiate duplication, we turned our attention to the recruitment of Sas6 at the site of PCL formation. To this end, we followed the distribution of Sas6 on round-, leaf-, and needle-shaped nuclear stages of spermatid cyst development. In the control *Ana1^OE* ninefold symmetrical centrioles of spermatids expressing Sas6-GFP, a prominent punctum of staining was apparent at the proximal tip at the round-shaped nuclear stage. This proximal tip staining had faded by the leaf-shaped nuclear stage, at which time a punctum of Sas6-GFP corresponding to the PCL predominated in lateral association with the proximal region of the centriole. This single punctum became individualized as part of a PCL at the needle-shaped nuclear stage (Fig. 6d). Similar observations on *rcd4²*;*Ana1^OE* spermatids showed Sas6-GFP associated with multiple splayed proximal ends of individual centrioles at the round nuclear stage. This resolved into a single punctum of laterally associated Sas6-GFP resembling PCL staining at the leaf nuclear stage. These puncta then dissociated into individualized

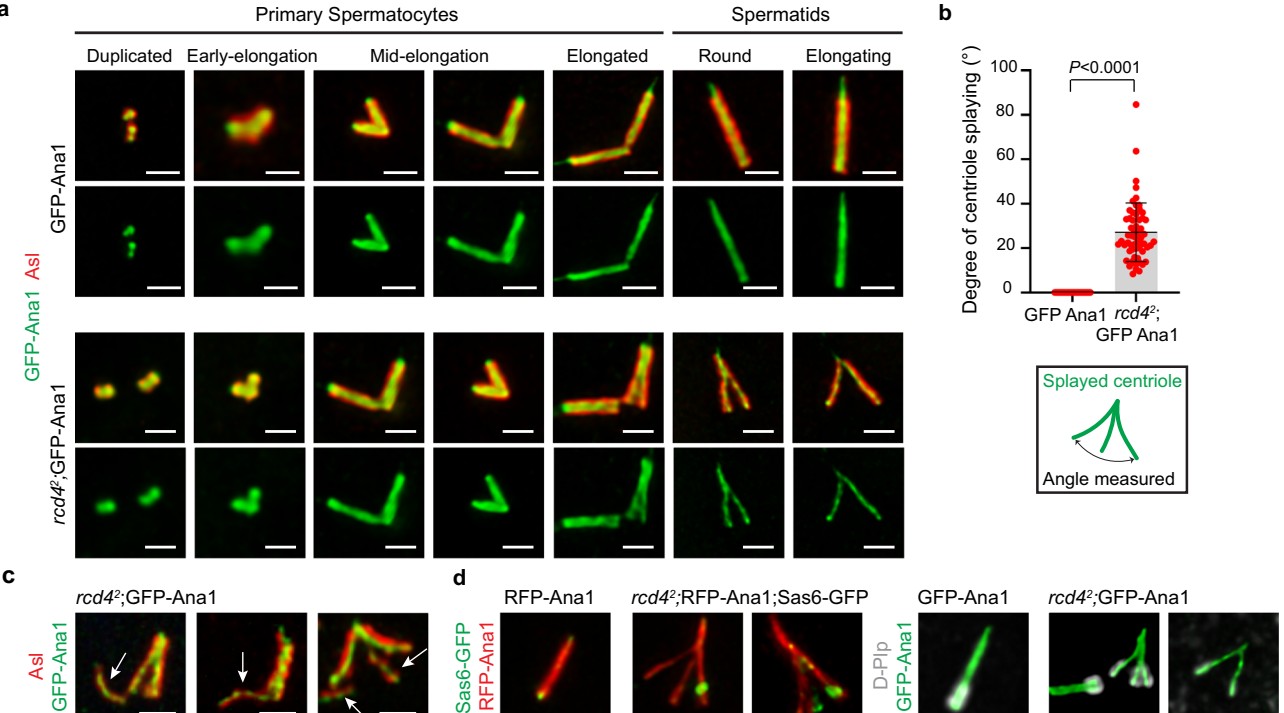

**Fig. 5 | Centriole splaying is initiated at the proximal end of the centriole. a** GFP-Ana1 (*Ana1^OE*) and *rcd4²*;GFP-Ana1 (*rcd4²; Ana1^OE*) primary spermatocytes (at centriole duplication, early elongation, mid-elongation, and elongated stages) and spermatids (at the round and elongating stages). Centrioles in the cells at the various stages were stained to reveal GFP-Ana1 (green) and Asl (red). Scale bar, 1 μm. **b** Degree of splaying measured in *rcd4²*; GFP-Ana1 spermatid centrioles. The angle between the two furthermost ends of the splayed centriole using GFP-Ana1 as a marker is presented. A two-tailed, unpaired *t* test was performed with a *P*-value of *P* < 0.0001. Mean values for GFP-Ana1 and *rcd4²*; GFP-Ana1 splayed centrioles are 0° and 27.16°, respectively. Mean values and SDs are shown for N = 60 splayed centrioles analyzed from 3 individual testes. Source data are provided as a Source Data file. **c** Examples of *rcd4²*; GFP-Ana1 centrioles in the process of splaying. Arrows show centriole fragments disassociating from the splaying centriole. Centrioles are immuno-stained to reveal GFP-Ana1 (green) and Asl (red). Scale bar, 1 μm. The experiment was repeated more than three times with similar results. **d** WT and *rcd4²*; RFP/GFP-Ana1 splayed centrioles stained to reveal Sas6-GFP (green) and D-Plp (gray). Scale bar, 1 μm. The experiment was repeated more than three times with similar results.

bodies present on each centriole in the cyst at the needle nuclear stage (Fig. 6d). This suggests the possibility that just as procentriole formation is initiated by association of the Plk4:Ana2:Sas6 complex with multiple sets of mother centriole microtubules (to give a structure like beads on a necklace) that then resolves into a single dot, similar events could take place on the splayed proximal ends of microtubules in the *rcd4²* spermatid following Ana1 overexpression.

## Discussion

In summary, our current findings offer additional functions for the conserved Ana3:Rcd4 complex other than those we described in the somatic cells of *Drosophila*, where it is required for the recruitment of Ana1, an important step in the process of centriole to centrosome conversion. The loss of Rcd4 led to cell division defects in the embryo and in the adult femoral chordotonal organ lineages resulting in abnormalities of the neurosensory cilia required for coordination. In the present study, we address differing aspects of *rcd4* function in the male germline in which centrioles have triplet microtubules, in contrast to the doublets of the soma, and these have to elongate in primary spermatocytes to form the giant centrioles that will become the basal body of functional sperm. The germline defects we describe appear to arise in the mitotic divisions of spermatogonia of *rcd4*-null males, and although Rcd4's partner protein Ana3 is initially recruited to the residual centrioles, it is subsequently lost upon centriole elongation in the extended G2 phase of primary spermatocytes. Thus, although not required to recruit Ana3, Rcd4 is required to maintain Ana3 at the centriole in the extended G2 phase that precedes meiosis.

Strikingly, 12.3% of the residual centrioles have a skinny appearance in the light microscope that is further revealed by electron microscopy to reflect a loss of the majority of triplet microtubules leaving the remaining microtubules as doublets or singlets. Thus, not only is the integrity of A–C linkages lost between triplets, but also the intra-integrity of the triplets themselves is forfeited.

The loss of Rcd4 and subsequently Ana3 have no effect upon the ability of either wild-type-like or skinny *rcd4*-null mutant centrioles to elongate. Moreover, both types of centrioles can associate with other key centriole components. The singlet or doublet microtubules found in skinny centrioles are, for example, able to accumulate components of the centriole cartwheel, Sas6, Gorab, and Cep135 at their putative proximal ends. Whether a complete cartwheel accumulates here, or whether it is a fragment of the cartwheel, is not clear, although the ninefold symmetry of centriole microtubules has been lost. Sas4 also accumulates at this putative proximal end, which serves as a focal point for the accumulation of D-Plp prior to meiotic entry just as with normal centrioles[19]. Finally, both the Ana1 and Asl proteins are distributed along the full length of both normal-like and skinny centrioles in *rcd4*-null spermatocytes. This longitudinal accumulation of Ana1 in a region devoid of Cep135 and in the absence of the Ana3:Rcd4 complex indicates that there must be an additional mechanism of Ana1 recruitment in centriole elongation that is independent of the Cep135 or Ana3-dependent mechanisms described to date. Taken together, these findings indicate that the ninefold symmetry of the centriole structure is not necessary for the association of the great majority of key centriole components in their correct spatial positions along the length of the centriole.

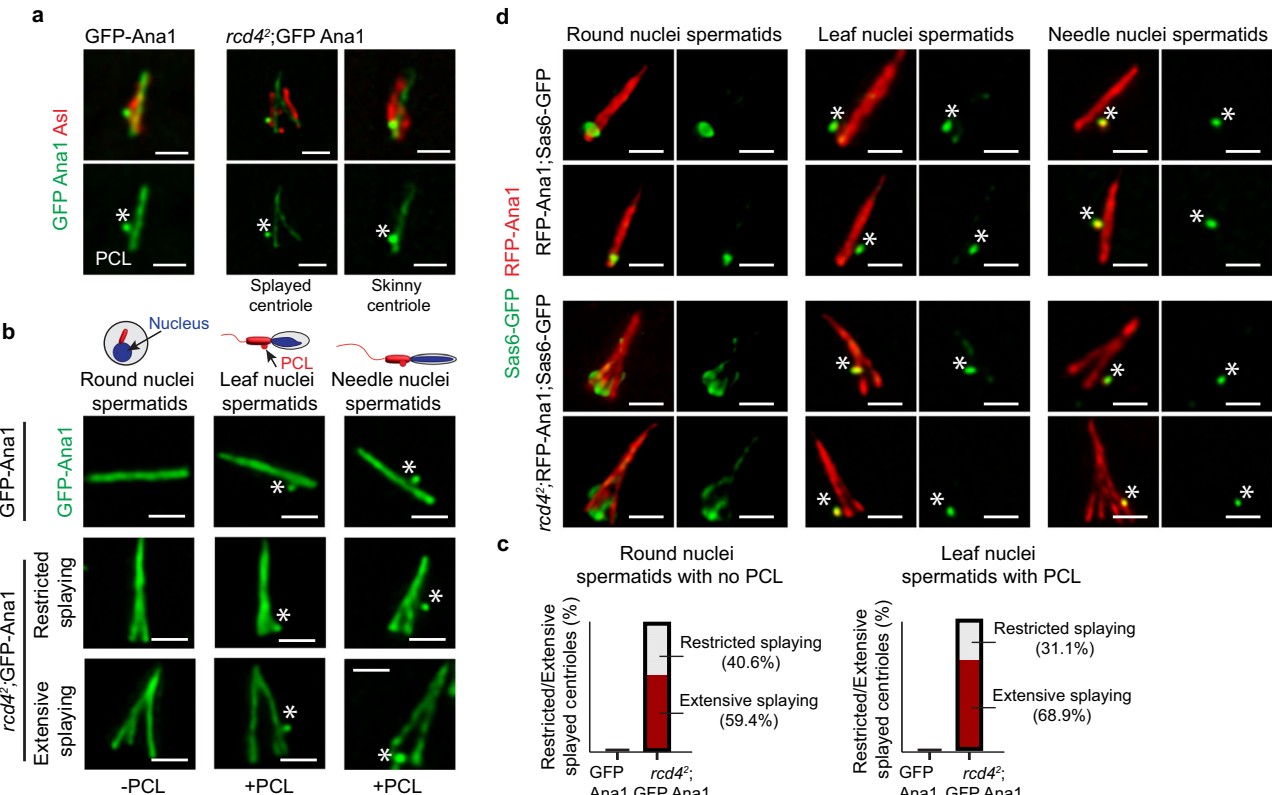

**Fig. 6 | Skinny and splayed spermatid centrioles are capable of initiating PCL formation. a** GFP-Ana1 (*Ana1^OE*) and *rcd4²*; GFP-Ana1 (*rcd4²*; *Ana1^OE*) spermatid centrioles immuno-stained to reveal GFP-Ana1 (green) and Asl (red). Initiation of Asl reduction in preparation for spermiogenesis results in observed fragmented Asl signal. GFP-Ana1 marks the PCL (proximal centriole-like structure; marked with an asterisk) alongside GFP-Ana1 centriole and *rcd4²*; GFP-Ana1 splayed and skinny centrioles. Scale bar, 1 μm. The experiment was repeated three times with similar results. **b** GFP-Ana1 and *rcd4²*; GFP-Ana1 spermatid centrioles reveal GFP-Ana1 centrioles (green) at successive round-, leaf-, and needle-shaped nuclear stages. Asterisks mark the GFP-Ana1 positive PCL. Examples of restricted and extensive centriole splaying are shown for *rcd4²*; GFP-Ana1 at each of the spermatid stages.

Scale bar, 1 μm. The experiment was repeated three times with similar results. **c** The proportion of splayed centrioles in GFP-Ana1 and *rcd4²*; GFP-Ana1 spermatids displaying restricted/extensive splaying at the round (N = 65 for GFP-Ana1 and N = 69 for *rcd4²*; GFP-Ana1 analyzed from 5 individual testes) and leaf (N = 60 for GFP-Ana1 and N = 61 for *rcd4²*; GFP-Ana1 analyzed from 5 individual testes) nuclear stages. The experiment was repeated two times with similar results. Source data are provided as a Source Data file. **d** Distribution of Sas6-GFP in round-, leaf-, and needle-like stage spermatid centrioles. Two examples are provided for RFP-Ana1; Sas6-GFP (*Ana1^OE*) and *rcd4²*; RFP-Ana1; Sas6-GFP (*rcd4²*; *Ana1^OE*) at each spermatid stage revealing Ana1 (red) and Sas6 (green). Asterisks mark the PCL. Scale bar, 1 μm. The experiment was repeated more than three times with similar results.

Centrioles lacking Rcd4 that have normal or skinny appearance can be induced to elongate further by elevated expression of Ana1 just as it occurs in otherwise wild-type centrioles[18]. We infer that this results in mechanical stress upon the *rcd4*-mutant centrioles of normal girth because these elongated structures exhibit splaying at their proximal ends. Electron microscopy reveals that such splaying results in a range of centriole morphologies: these include structures that retain triplet microtubules but come apart at their proximal ends. This indicates a requirement for the Ana3:Rcd4 complex to maintain the microtubule A–C linkages. As there are also structures in which triplets are reduced to doublet and singlet microtubules that splay apart at their putative proximal ends, it implies that the Ana3:Rcd4 complex also contributes to regulating the intra-microtubule stability of doublets and triplets per se. In mammalian cells, delta and epsilon tubulin are essential for the stability of triplet microtubules and in their absence, centrioles disintegrate during M/G1 phases and form de novo in each cell cycle, a phenotype that can be rescued by the microtubule-stabilizing drug taxol[20]. As *Drosophila* lacks delta and epsilon tubulin, we considered whether the disrupted triplet microtubules that form in *rcd4* spermatocytes and spermatids could also be stabilized by taxol. However, a 24 h treatment of cultured testes with taxol had no effect on the extent of splaying. This suggests that microtubule depolymerization is most likely not a contributing factor to the observed *rcd4*-mutant structural abnormalities. Several of the features of Rcd4 deficient centrioles are

similar to the consequences of extensive depletion of CPAP, the mammalian homolog of Sas4. CPAP-depleted centrioles also have incomplete triplet microtubules that fragment at their proximal ends due to a lack of cohesion between microtubule blades[21].

The loss of centrioles seen in *rcd4*-null spermatocytes could arise either because structural defects lead to a failure of the duplication cycle or because defects in the initiation of procentriole formation at the onset of duplication lead to the propagation of defective centrioles. Further studies are required to distinguish these possibilities. There is no way of knowing whether skinny centrioles can serve as mothers in the spermatogonium because *Drosophila* centrioles lack distal and sub-distal appendages, which are found on the mother centriole in mammalian cells, and centrobin, a daughter-specific molecule in centrioles of the soma, is present on all four spermatocyte centrioles. It would seem, however, that splayed and skinny centrioles in *rcd4*-null spermatids do have the potential to host the formation of a daughter centriole-like structure in the form of a PCL. Sas6-containing structures are initially present at multiple proximal ends of splayed *rcd4²*;*Ana1^OE* centrioles that are then replaced by a lateral focus of Sas6 against the proximal end of one of the splayed units that becomes an individualized PCL. This raises the question of whether the Plk4:Ana2 complex, which initiates centriole duplication, might be initially recruited to all splayed ends and then become concentrated on one of them in a manner analogous to the ring-to-dot

conversion seen in somatic cells[4]. Alternatively, might there be sufficient residual Sas6 at one of the splayed ends, which together with its associated microtubule and other centriolar components could provide a platform for PCL formation. Future studies to examine the recruitment of Plk4 and Ana2 should help distinguish these possibilities and their implications for models of how the symmetry of Plk4:Ana2 distribution is broken to establish a single site for daughter centriole formation.

## Methods

### Fly stocks and husbandry

All *Drosophila melanogaster* fly stocks were maintained at 25 °C, 60–80% humidity and on standard corn meal-yeast-sucrose media. w[1118] flies were used as wild-type controls and all mutant and transgenic flies were generated in a w[1118] strain genetic background.

The specific fly stocks used in this study are listed in Supplementary Table 1.

### Testes dissection and immunostaining

Testes were dissected from pharate adult males in PBS and transferred to a drop of PBS containing 5%-glycerol on a microscope slide. The testes were then squashed with a coverslip, snap-frozen in liquid nitrogen and fixed in chilled methanol for 3 min. The samples were rehydrated in PBS for 5 min, washed in PBS containing 0.01% Triton X-100 (PBSTx) for 10 min and incubated overnight in a humid chamber at 4 °C with the following primary antibodies: rabbit anti-Asl[22] (1:16,000; recognizes the N-terminus of the protein), chicken anti-D-Plp[23] (1:1000), rabbit anti-Ana1[13] (1:5000; recognizes the C-terminus of the protein) and guinea pig anti-Cep135 (1:500). The slides were washed three times in PBSTx for 10 min, incubated with appropriate secondary antibodies at 1:500 dilution: goat anti-chicken IgY Alexa Fluor 488 (Catalog # A-11039), goat anti-guinea pig IgG Alexa Fluor 488 (Catalog # A-11073), goat anti-rabbit IgG Alexa Fluor 568 (Catalog # A-11011), goat anti-chicken IgY Alexa Fluor 647 (Catalog # A32933) from Life Technologies, in a humid chamber for 1 h at 25 °C, followed by three 10 min washes in PBSTx before mounting in Vectashield containing DAPI.

### Confocal microscopy

Microscopic images were collected on a Leica Stellaris 8 Falcon laser scanning confocal microscope (Application Suite X software, LAS-X, Leica) using 63× (oil) objective, illuminated with a white light laser and detected by power Hybrid detectors. Images were deconvolved using in-built Leica Lightening software. All processing and analysis of microscope images were performed with ImageJ (ImageJ Fiji 1.53t, National Institutes of Health).

### Centriole length measurements

Only centrioles oriented perpendicular to the imaging axis were measured to avoid ambiguity contributed by tilting. System-optimized Z-stack steps, spanning the entire centriole volume were taken on the Leica Stellaris 8 Falcon confocal laser scanning microscope, as described above. The length of the centrioles was measured using the line profile tool in ImageJ (ImageJ Fiji 1.53t, NIH) and statistically analyzed.

### Splayed centriole angle measurements

The degree of splaying was measured as the angle between the two furthermost ends of the splayed centriole using GFP-Ana1 as a marker. The angle was measured using the angle tool in ImageJ (ImageJ Fiji 1.54f, NIH) and statistically analyzed.

### Statistical analysis

The population size (N), sample size (n), mean and SD are indicated in corresponding figures and figure legends. Data collected for quantification purposes were analyzed with a two-tailed, unpaired *t* test by GraphPad Prism (version 10.2.0 (392)). *P*-Values for each analysis are indicated in the corresponding figures and figure legends; a 95% confidence interval was applied in all statistical tests.

### Electron microscopy

*Drosophila* testes were dissected into a cold (4 °C) fixative solution of 3% glutaraldehyde, 1% paraformaldehyde, 5% sucrose in 0.1 M sodium cacodylate trihydrate. Testes were rinsed with cacodylate buffer containing 10% Ficoll (an extracellular cryoprotectant) and placed individually into brass high-pressure freezing planchettes (Ted Pella, Inc. Redding CA) prefilled with the same buffer.

Samples were ultra-rapidly frozen with a HPM-010 high-pressure freezing machine (Bal-Tec AG, Balzers, Liechtenstein) and transferred to liquid nitrogen. Planchettes containing vitrified tissue were transferred under liquid nitrogen to cryotubes (Nunc) prefilled with 2% $OsO_4$, and 0.05% uranyl acetate in acetone. Samples were placed in a AFS-2 freeze-substitution machine (Leica Microsystems, Vienna), processed at −90 °C for 72 h, warmed to −20 °C over 12 h, and further processed at that temperature for 24 h, then rinsed with acetone and flat-embedded in Epon-Araldite resin (Electron Microscopy Sciences, Port Washington PA) between Teflon-coated glass slides. The resin was polymerized at 60° for 48 h.

Embedded tissue blocks were observed by phase-contrast microscopy to select well-preserved and optimally oriented regions, then extracted with a scalpel and glued to plastic sectioning stubs. Semi-thick (170 nm) serial sections were cut with a EM UC-6 ultramicrotome (Leica Microsystems) using a diamond knife (Diatome US). Sections were collected onto Formvar-coated copper/rhodium slot grids and stained with 3% uranyl acetate and lead citrate. Colloidal gold particles (10 nm) were placed on both surfaces of the grids to serve as fiducial markers for tomographic image alignment.

### Dual-axis tomography

Grids were placed in a dual-axis tomography holder (Model 2040, E.A. Fischione Instruments, Inc., Export, PA) and imaged with a Tecnai-12 transmission electron microscope (Thermo-Fisher Scientific) at 120KeV. Centrioles were located and tracked manually across serial sections. For tomography, grids were tilted +/−62° and images acquired at 1° increments. The grid was then rotated 90° and a similar tilt-series was taken about the orthogonal axis. Tilt-series were acquired automatically using the SerialEM software package[24] and recorded with a CCD camera (US1000, Gatan, Inc.). Tomographic datasets were processed, analyzed and segmented with the IMOD software package[25,26] on iMac Pro and Mac Studio M1 computers (Apple, Inc.).

### Reporting summary

Further information on research design is available in the Nature Portfolio Reporting Summary linked to this article.

## Data availability

Source data are provided with this paper.

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

## Acknowledgements

We are grateful to Dr. Andres Collazo and Dr. Giada Spigolon of the Biological Imaging Facility at Caltech for their support and to Caltech Beckman Institute Cryo-EM Facility for the use of the Tecnai electron microscope. We are thankful to Paula Coelho, Levente Kovacs, Agota Nagy, and Sarolta Szentes for many discussions. This work was supported by a grant from the National Institute of Neurological Disorders and Stroke of the National Institutes of Health under award no. R01NS113930.

## Author contributions

P.P contributed to the conception of the study, designed experiments, performed all *Drosophila* genetics and fluorescence microscopy studies, carried out data analysis and wrote the manuscript; M.S.L prepared samples for and performed all electron tomography studies; D.M.G contributed to the conception and supervision of the study, and in writing the manuscript.

## Competing interests

The authors declare no competing interests.
