## [Peer Review File · Nature Communications]

9-fold symmetry is not essential for centriole elongation and formation of new centriole-like structuresReviewer #1 (Remarks to the Author):

In this manuscript Glover and colleagues study the consequences of Rcd4 (PPP1R35 in humans) loss in *Drosophila* spermatocytes. In somatic cells, newly formed centrioles need to be converted into centrosomes during G2/M. As the authors have already demonstrated in flies (and others in mammalian cells), this process involves recruitment of Ana1/CEP295 by Ana3/RTTN and Rcd4/PPP1R35. They have also shown that Rcd4 is required for centriole duplication cycles in embryos, somatic cells and the male germline (~30% reduction in centrioles in latter). Building on these previous studies here the authors investigate the function of Rcd4 in assembly and maintenance of the atypical centrioles found in the *Drosophila* male germline.

This paper reveals that in addition to causing an overall reduction in centriole numbers, Rcd4 deletion in sperm results in fragmentation of triplet microtubules into singlet and doublet microtubules, leading to partial loss of centriole walls and the appearance of 'skinny' centrioles. Such skinny centrioles still elongate and recruit proximal end markers. The authors further test structural resilience of these residual Rcd4 mutant centrioles by overexpressing Ana1, which causes excess elongation. Whereas in wild type spermatids, centrioles appear longer but otherwise normal, Rcd4 mutant centrioles exhibit extensive splaying at their proximal end, a phenotype visible in spermatids. Based on these observations the authors propose that Rcd4 facilitates A-C microtubule links between triplets as well as intra-microtubule stability within triplets and doublets, thus contributing to the characteristic ninefold symmetry of centrioles. Because these abnormal centriole structures still allow elongation and formation of a proximal centriole like structure (PCL), the authors conclude that the ninefold symmetry is dispensable for both centriole elongation and initiation of duplication, a point made in the title as well as the abstract.

The manuscript is well written, and data quality and presentation are both excellent. My main criticisms are that 1) this is a largely descriptive study perhaps because the nature of the model system prevents a more mechanistic approach and 2) it provides limited additional insight over previous papers from the group (and others), aside from some intriguing tissue- and possibly species-dependent differences such as Rcd4 dependency of Ana3 maintenance on centrioles in sperm but not in somatic cells. While I do find the phenotypes interesting, I would expect a more in-depth functional characterisation of Rcd4 and Ana3 function in microtubule stability. Also, the conclusion that centriole duplication can be initiated without the 9-fold symmetry of centrioles is currently not fully supported by the data.

Specific comments:

1. The authors suggest that centriole duplication can be initiated by a centriole lacking the ninefold symmetry. There are two issues with this statement. First, with the experimental approach here it seems difficult to exclude the possibility that the centriole still had triplet microtubules arranged with 9-fold symmetry when the PCL formed with the fragmentation/splaying occurring subsequently. Second, PCL is a highly atypical procentriole; while it relies on PLK4/SAK, Sas5 and Sas6 for its formation, it does not need Sas4. PCL consists of a central tube and an electron-dense wall (the latter may disappear in spermatids) it lacks a cartwheel, and is more akin to a procentriole arrested at the earliest stage of its assembly. Therefore, despite sharing some similarities, PCL assembly may not be analogous to centriole duplication.

Based on Fig 4d, the cartwheel remains relatively intact in 'early' splayed centrioles, breaking up more in 'advanced' splayed centrioles, and so I wonder if it made sense to score presence/absence of PCL according to degree of splaying in late spermatids. If this is a gradual process, and all early splayed centrioles have a PCL already, the statements about the ninefold symmetry and centriole duplication should be adjusted.

The authors may also want to comment on why there seems to be always only one PCL per splayed group of microtubules. If this is so, it suggests that PCL assembles prior to splaying, or why would not there be multiple PCLs per group? Judged from the data in the current manuscript, it appears that the wall remnants with the cartwheel are still the preferred site for assembly.

2. *Drosophila* sperm centrioles are atypical in many ways including their sensitivity to microtubule polymerising and depolymerising drugs. Furthermore, delta and epsilon tubulin are absent from *Drosophila*, whereas they play an important role in mammalian centrioles. Indeed, their absence

causes a reduction in outer diameter of centrioles due to centrioles lacking triplet microtubules (Wang et al., eLife, 2017). Such centrioles disintegrate during M/G1 phases and form de novo in each cell cycle, suggesting that delta and epsilon tubulin are vital for stability of microtubule triplets. Intriguingly, this phenotype could be rescued by taxol. It would be useful to comment on whether microtubule triplet stability in *Drosophila* sperm requires a different stabilising mechanism. Furthermore, if technically feasible, a similar experiment with taxol in Rcd4 mutants would be informative.

Reviewer #2 (Remarks to the Author):

In this manuscript, Panda et al describe a new, hidden function for a known centriolar protein called Rcd4/PPP1R35, which is revealed in *Drosophila* male germline. It has been reported previously by several groups that in somatic cycling cells, Rcd4/PPP1R35 promotes the recruitment of several key molecules to newborn centrioles, including Ana1/CEP295 and Asl/CEP152, which eventually enables centriole-to-centrosome conversion. That is, in somatic cycling cells depleted of Rcd4/PPP1R35, procentrioles can still form, but they lack Ana1 or Asl and are structurally unstable during cell cycle and cannot be maintained as centrosomes. Here, in *Drosophila* male germline, the authors found that Rcd4/PPP1R35 is surprisingly not essential for the recruitment of Ana1/CEP295 and Asl/CEP152, a notable condition allowing Rcd4-null centrioles to be maintained in cells for further examination. Structural analyses revealed that a fraction of Rcd4-null centrioles display an unusually skinny shape, with triplet microtubule blades fragmentizing into a reduced number of doublet or singlet microtubules, thus losing the typical 9-fold symmetric architecture. In addition, Rcd4-null centrioles of normal shape were also shown to be structurally unstable, as they splay at their proximal ends when induced to elongate. These skinny or splay centrioles were further shown to be capable of recruiting other centriolar components, undergoing elongation, or even initiating the loading of an assembly site for the next run of centriole duplication. Together, the authors conclude that intact 9-fold symmetry of microtubule triplets is not essential for many aspects of centriole growth and development.

This study is well done – the data is clear and of high quality, and the conclusion is largely supported. I have the following questions/suggestions:

1. The skinny centriole, its origin and its relationship with the normal-looking centriole that it pairs with. It seems that the skinny centriole could be a breakdown/splayed product from an intact Rcd4-null centriole, which gets stabilized (temporarily). In the case of paired skinny, is it possible to determine if the skinny is the daughter, mother, or the splayed product of the nearby centriole that appears intact (and may miss just one chunk of a triplet or an entire triplet)? Is there a case in paired skinny where the skinny is older than, or being the mother of, the intact centriole? I am not sure if there is a way to determine the mother-daughter centriole relationship in cells of *Drosophila* male germline.
2. Is there evidence showing that the skinny centriole is formed skinny at the start? If not, can we really argue that 9-fold symmetry as the way to build centrioles not essential for centriole growth and maintenance as suggested by the title? It seems clear to me that once centrioles are formed, the continuous maintenance of 9-fold symmetry is not essential for other aspects of centriole growth/development.
3. Can the author discuss why they think Rcd4/PPP1R35 behaves so differently in *Drosophila* male germline regarding its role in centriole to centrosome conversion?

Responses to Reviewer #1's comments

1. [Reviewer's comment]

"The manuscript is well written, and data quality and presentation are both excellent. My main criticisms are that 1) this is a largely descriptive study perhaps because the nature of the model system prevents a more mechanistic approach and 2) it provides limited additional insight over previous papers from the group (and others), aside from some intriguing tissue- and possibly species-dependent differences such as Rcd4 dependency of Ana3 maintenance on centrioles in sperm but not in somatic cells. While I do find the phenotypes interesting, I would expect a more in-depth functional characterisation of Rcd4 and Ana3 function in microtubule stability. Also, the conclusion that centriole duplication can be initiated without the 9-fold symmetry of centrioles is currently not fully supported by the data."

1. [Authors' response]

We thank the referee for their feedback on our work. Indeed, we have taken the experimental system almost as far as is possible and functional characterisation becomes difficult in the fly testes from this particular point onwards. That said, we feel that the findings are particularly striking and will be valuable for the community to consider as they raise possibilities of alternative explanations for some commonly held beliefs. We thank the referee for raising some important questions about centriole duplication and stability, which we have addressed as outlined below.

2. [Reviewer's comment]

"1. The authors suggest that centriole duplication can be initiated by a centriole lacking the ninefold symmetry. There are two issues with this statement. First, with the experimental approach here it seems difficult to exclude the possibility that the centriole still had triplet microtubules arranged with 9-fold symmetry when the PCL formed with the fragmentation/splaying occurring subsequently. Second, PCL is a highly atypical procentriole; while it relies on PLK4/SAK, Sas5 and Sas6 for its formation, it does not need Sas4. PCL consists of a central tube and an electron-dense wall (the latter may disappear in spermatids) it lacks a cartwheel and is more akin to a procentriole arrested at the earliest stage of its assembly. Therefore, despite sharing some similarities, PCL assembly may not be analogous to centriole duplication."

2. [Authors' response]

We thank the referee for raising these two points.

We apologise for not being clear in the first version of the paper that PCL formation begins synchronously at a particular stage of spermatid development. Thus, the 64 spermatids within a cyst form the PCL synchronously. We have now staged spermatids at the successive round-, leaf, and needle-shaped nuclear stages and show that the majority of centrioles show extensive splaying at the round nuclear stage before the PCL has formed. The PCL then appears synchronously at single sites on both splayed and non-splayed centrioles at the leaf nuclear stage (this is outlined in more detail below in response to point 3). This supports PCL assembly subsequent to splaying.

We also now show that Sas6 is present at multiple splayed proximal ends of splayed *rcd4²* centrioles, diminishes, and then re-appears on a single PCL structure. We discuss how this might resemble the ring-to-dot conversion of Plk4 in the initiation of conventional centriole duplication.

We acknowledge that the PCL is indeed a highly atypical centriole-like structure that is unique to the *Drosophila* male germline. However, in the context of this study and the tissue being analysed, the PCL is the natural product of centriole replication. Despite it lacking several centriole components leading to its structural divergence, the PCL is capable of recruiting PCM, forming an aster, and templating the formation of a daughter centriole upon fertilization in the *Drosophila* zygote¹. We have attempted to clarify the differences between the PCL and conventional centrioles throughout the manuscript and in particular we indicate in the summary paragraph that centrioles lacking 9-fold symmetry (skinny and splayed centrioles) display aspects of the initiation of centriole-like duplication rather than implying complete centriole duplication.

3. [Reviewer's comment]

"Based on Fig 4d, the cartwheel remains relatively intact in 'early' splayed centrioles, breaking up more in 'advanced' splayed centrioles, and so I wonder if it made sense to score presence/absence of PCL according to degree of splaying in late spermatids. If this is a gradual process, and all early splayed centrioles have a PCL already, the statements about the ninefold symmetry and centriole duplication should be adjusted."

3. [Authors' response]

We thank the reviewer for the suggestion and have addressed the point by providing some additional supporting data.

The onset of centriole splaying appears variable. However, it is difficult to establish an accurate time course of splaying from observations of fixed samples. We have therefore decided to refer to the degree of splaying as "restricted" or "extensive" rather than "early" or "advanced". That said, the splaying becomes obvious in late spermatocyte/round spermatid stages but there is a slight increase in the ratio of extensive: restricted splaying in later-staged spermatids. To clarify the formation of the PCL relative to the degree of splaying, we have now systematically staged spermatid development in relation to nuclear morphology (**updated Extended Data Fig. 4**) as a proxy for developmental time.

We include data from additional experiments as **Fig. 6b and 6c** and have modified the accompanying text as follows:

[Lines 175-190, p.6]

"The presence of PCL bodies near the proximal ends of both splayed and skinny centrioles in *rcd4² Ana1^{OE}* spermatids (Fig. 6a) led us to relate PCL formation to nuclear morphology, which changes as spermatids elongate (Fig. 6b; Extended Data Fig. 4). In control *Ana1^{OE}* spermatids, none of the centrioles were splayed at the round spermatid stage and they all lacked PCLs. PCLs appeared synchronously within the spermatid cyst at the leaf nuclear stage as evident by the GFP-Ana1 positive bulge in close proximity to one end of all centrioles. Distinct individualized PCLs were then seen within a cyst of spermatids at the needle-like stage (Fig. 6b; Extended Data Fig. 4). The proportion of splayed centrioles in *rcd4² Ana1^{OE}* spermatids showing extensive splaying at the round nuclear stage was 59.4% (n=41/69) whereas splaying was restricted in 40.6% (n=28/69) of the remaining splayed centrioles (Fig. 6c). None of these had an associated PCL indicating that the greater proportion of extensive centriole splaying takes place before PCL formation (Extended Data Fig. 4). The proportion of extensively splayed centrioles further increased to 68.9% (n=42/61) at the leaf-shaped nuclear stage when single PCLs appeared simultaneously on all centrioles. We also observed the formation of the PCL on skinny centrioles at these stages. Thus, PCLs can assemble on both skinny and splayed centrioles that are losing the connections between the 9-fold symmetrical microtubules."

4. [Reviewer's comment]

"The authors may also want to comment on why there seems to be always only one PCL per splayed group of microtubules. If this is so, it suggests that PCL assembles prior to splaying, or why would not there be multiple PCLs per group? Judged from the data in the current manuscript, it appears that the wall remnants with the cartwheel are still the preferred site for assembly."

4. [Authors' response]

To address this comment, we have carried out a more thorough examination of the recruitment of Sas6 and have included the data in **Fig. 6d** and added the following in the text:

[Lines 191-210, p.6-7]

"This led us to ask the extent to which, when present, 9-fold symmetrical units at the splayed proximal end of the centriole participated in the initiation of this form of centriole-like duplication to generate the PCL. At this stage of development, where we are currently not able to visualise the recruitment of Plk4 or Ana2, the molecules that initiate duplication, we turned our attention to the recruitment of Sas6 at the site of PCL formation. To this end, we followed the distribution of Sas6 on round-, leaf-, and needle-shaped nuclear stages of spermatid cyst development. In the control *Ana1^{OE}* 9-fold symmetrical centrioles of spermatids expressing Sas6-GFP, a prominent punctum of staining was apparent at the proximal tip at the round-shaped nuclear stage. This proximal tip staining had faded by the leaf-shaped nuclear stage, at which time a punctum of Sas6-GFP corresponding to the PCL predominated in lateral association with the proximal region of the centriole. This single punctum became individualized as part of a PCL at the needle-shaped nuclear stage (Fig. 6d). Similar observations on *rcd4² Ana1^{OE}* spermatids showed Sas6-GFP associated with multiple splayed proximal ends of individual centrioles at the round nuclear stage. This resolved into a single punctum of laterally associated Sas6-GFP resembling PCL staining at the leaf nuclear stage. These puncta then dissociated into individualized bodies present on each centriole in the cyst at the needle nuclear stage (Fig. 6d). This suggests the possibility that just as procentriole formation is initiated by association of the Plk4:Ana2:Sas6 complex with multiple sets of mother centriole microtubules (to give a structure like beads on a necklace) that then resolves into a single dot, similar events could take place on the splayed proximal ends of microtubules in the *rcd4²* spermatid following Ana1 overexpression."

We also included the following into the discussion section:

[Lines 275-285, p.9]

“Sas6-containing structures are initially present at multiple proximal ends of splayed *rcd4*² *Ana1*^{OE} centrioles that are then replaced by a lateral focus of Sas6 against the proximal end of one of the splayed units that becomes an individualized PCL. This raises the question of whether the Plk4:Ana2 complex, which initiates centriole duplication, might be initially recruited to all splayed ends then becomes concentrated on one of them in a manner analogous to the ring to dot conversion seen in somatic cells. Alternatively, might there be sufficient residual Sas6 at one of the splayed ends, which together with its associated microtubule and other centriolar components could provide a platform for PCL formation? Future studies to examine the recruitment of Plk4 and Ana2 should help distinguish these possibilities and their implications for models of how the symmetry of Plk4:Ana2 distribution is broken to establish a single site for daughter centriole formation.

5. [Reviewer’s comment]

“Drosophila sperm centrioles are atypical in many ways including their sensitivity to microtubule polymerising and depolymerising drugs. Furthermore, delta and epsilon tubulin are absent from Drosophila, whereas they play an important role in mammalian centrioles. Indeed, their absence causes a reduction in outer diameter of centrioles due to centrioles lacking triplet microtubules (Wang et al., eLife, 2017). Such centrioles disintegrate during M/G1 phases and form de novo in each cell cycle, suggesting that delta and epsilon tubulin are vital for stability of microtubule triplets. Intriguingly, this phenotype could be rescued by taxol. It would be useful to comment on whether microtubule triplet stability in Drosophila sperm requires a different stabilising mechanism. Furthermore, if technically feasible, a similar experiment with taxol in Rcd4 mutants would be informative.”

5. [Authors’ response]

We appreciate the reviewer’s input and suggestion to investigate the potential mechanism behind *rcd4*’s skinny and splayed mutant centrioles.

Not much is known in the field in terms of factors that contribute towards maintaining centriole microtubule stability in somatic and germline tissues of flies. Moreover, components such as zeta, epsilon and delta tubulins are known to be absent in *Drosophila*⁶.

In fact, early male germline centrioles are composed of a mixture of microtubule doublets and triplets and only in the spermatocyte stage do they become exclusively composed of microtubule triplets⁷. It is hypothesised that the spermatocyte centrioles acquire microtubule triplet characteristics as a means to provide additional structural stability to the extensively elongated giant centrioles.

Thus, not much is known yet about centriolar microtubule cohesion in the *Drosophila* male germline specifically, except that our data seems to suggest a role for Rcd4 and its partner Ana3 in contributing towards inter/intra-microtubule stability via a yet unknown mechanism.

As suggested by the reviewer, we performed a taxol-treatment experiment. GFP-Ana1 (control) and *rcd4*²; GFP-Ana1 testes were dissected and cultured in 5 μ M Taxol or DMSO for 24 hrs (efficacy of drug treatment was established in a previous study⁸). We then quantified the proportion of resulting splayed centrioles after taxol-treatment. If microtubules in *rcd4*-mutant centrioles are unstable and undergo depolymerisation giving rise to splayed centrioles, we would then expect to observe a reduced proportion of splayed centrioles after taxol treatment that stabilises microtubules. However, quantifications of the percentage of splayed centrioles in taxol-treated spermatids (61%, N=119) was not significantly different from the DMSO-treated spermatids (62.2%, N=261) (see below) suggesting that microtubule depolymerisation is most likely not a contributing factor to the observed *rcd4*-mutant centriolar structural abnormalities.

While we still do not yet know the basis of *rcd4*-mutant skinny and splayed centriole formation, our data indicates a requirement for the Rcd4:Ana3 complex to maintain the microtubule A-C linkages and also contribute towards regulating the intra-microtubule stability of triplets.

We have added the following point to the discussion section of the manuscript:

[Lines 254-261, p.8]

“In mammalian cells, delta and epsilon tubulin are essential for the stability of triplet microtubules and in their absence, centrioles disintegrate during M/G1 phases and form de novo in each cell cycle, a phenotype that can be rescued by the microtubule stabilising drug taxol. As *Drosophila* lacks delta and epsilon tubulin, we considered whether the disrupted triplet microtubules that form in *rcd4* spermatocytes and spermatids could also be stabilised by taxol. However, a 24hrs treatment of cultured testes with taxol had no effect on the extent of splaying (data not shown). This suggests that microtubule depolymerisation is most likely not a contributing factor to the observed *rcd4*-mutant structural abnormalities.”

Responses to Reviewer #2's comments

1. [Reviewer's comment]

“This study is well done – the data is clear and of high quality, and the conclusion is largely supported.”

1. [Authors' response]

We appreciate the reviewer's positive feedback and have tried to address the questions raised.

2. [Reviewer's comment]

*“1. The skinny centriole, its origin and its relationship with the normal-looking centriole that it pairs with. It seems that the skinny centriole could be a breakdown/splayed product from an intact *Rcd4*-null centriole, which gets stabilized (temporarily). In the case of paired skinny, is it possible to determine if the skinny is the daughter, mother, or the splayed product of the nearby centriole that appears intact (and may miss just one chunk of a triplet or an entire triplet)? Is there a case in paired skinny where the skinny is older than, or being the mother of, the intact centriole? I am not sure if there is a way to determine the mother-daughter centriole relationship in cells of *Drosophila* male germline.”*

2. [Authors' response]

We agree that the unpaired skinny centrioles could very well be stabilised broken fragments of the *rcd4*-null mutant centrioles. However, we have no proper means of experimentally verifying whether this is the case.

Unlike mammalian centrioles, *Drosophila* centrioles do not have distal/sub-distal appendages making it difficult to distinguish between mother and daughter centrioles. Although Centrobin⁹, a daughter-centriole specific protein, confers daughter identity to centrioles in the *Drosophila* larval neuroblasts¹⁰, sensory neurons¹¹, and ommatid cells¹², it does not discriminate between mother-daughter centrioles in the primary spermatocytes of the male germ line. Centrobin is found at the proximal region of both paired centrioles in spermatocytes¹³.

We address the question of mother-daughter relationship by adding the following sentence in the text:

[Lines 269-273, p.9]

“There is no way of knowing whether skinny centrioles can serve as mothers in the spermatogonium because *Drosophila* centrioles lack distal and sub-distal appendages, which are found on the mother centriole in mammalian cells, and centrobin, a daughter specific molecule in centrioles of the soma, is present on all four spermatocyte centrioles.”

We were equally keen on investigating the identify of the paired skinny centrioles but are limited due to the above reasons. Nevertheless, we strongly believe that the skinny centriole when paired with a wild-type-like centriole is not merely a broken/splayed product of the intact centriole but is a structurally independent centriole. This can be supported by our electron tomogram of a paired skinny centriole in Fig. 1d where a fully intact 9 microtubule-triplet centriole is orthogonally associated with a cluster of three-microtubules representing the skinny centriole. Moreover, consistently in all our confocal images, the paired normal: skinny centrioles always adopt the V-shaped configuration unique to spermatocytes. If the skinny centrioles were indeed broken/splayed fragments, we would expect to see them in varying conformations (as exemplified in Fig. 5c) relative to the intact centriole.

We have added the following sentence in the manuscript:

[Lines 60-62, p.2]

“The pairing of a skinny centriole orthogonal to one of wild-type appearance suggests that the former is not derived by breakdown of the latter but that it arose independently in the current or previous cell generation.”

3. [Reviewer’s comment]

“2. Is there evidence showing that the skinny centriole is formed skinny at the start? If not, can we really argue that 9-fold symmetry as the way to build centrioles not essential for centriole growth and maintenance as suggested by the title? It seems clear to me that once centrioles are formed, the continuous maintenance of 9-fold symmetry is not essential for other aspects of centriole growth/development.”

3. [Authors’ response]

Centrioles in the early spermatogonial cells are structurally short resembling centrioles found in the somatic tissues of the fly. These centrioles appear as concentric rings when stained for Asl and D-Plp (WT centrioles in Fig. 2b and 2c). In the *rcd4²* spermatogonium, we observed morphologically aberrant Asl and D-Plp positive structures (Fig. 2b and 2c) in addition to the expected concentric rings of Asl and D-Plp. As these structures are rather small, it was difficult to obtain higher resolution images with regular confocal microscopy and was almost impossible to find and identify them via electron tomography. Moreover, we observed similar structures in *rcd4²* early primary spermatocytes (duplicated panel in Fig. 2c) - mini skinny centrioles associated with normal ring-like centrioles even before centriole elongation has begun. We would expect such pairs of centrioles to give rise to the paired skinny centrioles we observe in mature primary spermatocytes. Taken together, this suggests that the skinny centrioles do indeed begin as structurally abnormal centrioles early in spermatogenesis rather than being the broken/splayed by-product of existing intact mutant centrioles.

The relevant text (added in *italics*) states:

[Lines 86-91, p.3]

“Thus, the structural consequences of the absence of Rcd4 protein are manifested in the spermatogonial divisions as centriole loss and generation of skinny centrioles *giving rise to spermatocytes having normal and skinny centrioles*. Together with centrioles of normal appearance, skinny centrioles are able to undergo elongation in the spermatocyte’s

extended G2 phase and persist through the meiotic divisions segregating into spermatids (Fig. 2c).”

4. [Reviewer’s comment]

“3. Can the author discuss why they think Rcd4/PPP1R35 behaves so differently in *Drosophila* male germline regarding its role in centriole to centrosome conversion?”

4. [Authors’ response]

We thank the reviewer for raising this question and have now addressed it in the manuscript.

The centrioles of the *Drosophila* male germline are unique in several respects compared to centrioles at other developmental stages: they develop triplet microtubules; they become excessively long; and once they form the sperm basal body, they direct formation of a daughter centriole-like structure, the PCL. Our demonstration of a role for Rcd4 in the maintenance of centriolar inter/intra-microtubule stability might reflect a function unique to the distinct properties of male germline centrioles that might be unrelated to our previous demonstration of a role for Rcd4 and its partner Ana3 in Ana1 recruitment and centriole to centrosome conversion in somatic cells.

As Cep135 is also required for Ana1 recruitment for centriole to centrosome conversion¹⁴ in somatic cells¹⁴, it is possible that the Cep135-Ana1-Asl network accounts for the recruitment of Ana1 and Asl in the absence of Rcd4 and Ana3 in male germline centrioles. Alternatively, the initial loading of Ana3 onto the centrioles in the spermatogonia (Fig. 3c) could be sufficient to recruit Ana1 and Asl. The extension of Ana1 and Asl along the length of the giant centriole would appear to represent a third method of Ana1 recruitment.

We have added the following sentence in the text:

[Lines 238-241, p.8]

“This longitudinal accumulation of Ana1 in a region devoid of Cep135 and in the absence of the Ana3:Rcd4 complex indicates that there must be an additional mechanism of Ana1 recruitment in centriole elongation that is independent of the Cep135 or Ana3 dependent mechanisms described to date.”

Additional References:

1. Blachon, S., Khire, A. & Avidor-Reiss, T. The origin of the second centriole in the zygote of *Drosophila melanogaster*. *Genetics* **197**, 199–205 (2014).
2. Sonnen, K. F., Schermelleh, L., Leonhardt, H. & Nigg, E. A. 3D-structured illumination microscopy provides novel insight into architecture of human centrosomes. *Biol Open* **1**, 965–976 (2012).
3. Arquint, C. *et al.* STIL binding to Polo-box 3 of PLK4 regulates centriole duplication. *Elife* **4**, (2015).
4. Ohta, M. *et al.* Direct interaction of Plk4 with STIL ensures formation of a single procentriole per parental centriole. *Nat Commun* **5**, 5267 (2014).
5. Kim, T. S. *et al.* Hierarchical recruitment of Plk4 and regulation of centriole biogenesis by two centrosomal scaffolds, Cep192 and Cep152. *Proc Natl Acad Sci U S A* **110**, (2013).
6. Turk, E. *et al.* Zeta-Tubulin Is a Member of a Conserved Tubulin Module and Is a Component of the Centriolar Basal Foot in Multiciliated Cells In Brief. *Current Biology* **25**, (2015).
7. Gottardo, M., Callaini, G. & Riparbelli, M. G. The *Drosophila* centriole – conversion of doublets into triplets within the stem cell niche. *J Cell Sci* **128**, 2437–2442 (2015).
8. Riparbelli, M. G., Cabrera, O. A., Callaini, G. & Megraw, T. L. Unique properties of *Drosophila* spermatocyte primary cilia. *Biol Open* **2**, 1137–1147.
9. Zou, C. *et al.* Centrobin: a novel daughter centriole-associated protein that is required for centriole duplication. *J Cell Biol* **171**, 437–45 (2005).
10. Januschke, J. *et al.* Centrobin controls mother-daughter centriole asymmetry in *Drosophila* neuroblasts. (2013) doi:10.1038/ncb2671.
11. Gottardo, M. *et al.* Loss of Centrobin Enables Daughter Centrioles to Form Sensory Cilia in *Drosophila*. *Current Biology* **25**, 2319–2324 (2015).
12. Riparbelli, M. G., Persico, V., Gottardo, M. & Callaini, G. The developing *Drosophila* eye - a new model to study centriole reduction. *J Cell Sci* **131**, (2018).
13. Reina, J. *et al.* Centrobin is essential for C-tubule assembly and flagellum development in *Drosophila melanogaster* spermatogenesis. *J Cell Biol* **217**, 2365–2372 (2018).
14. Fu, J. *et al.* Conserved molecular interactions in centriole-to-centrosome conversion. *Nat Cell Biol* **18**, 87–99 (2016).

Reviewer #1 (Remarks to the Author):

The authors have satisfactorily addressed the points I raised and substantially improved their manuscript. I am pleased to be able to recommend their work for publication.

Reviewer #2 (Remarks to the Author):

The authors have addressed my concerns. Overall, this is a solid study with interesting results that raise new ideas about centriole biogenesis previously not considered. I therefore fully support the publication.